# Destruction of Polyelectrolyte Microcapsules Formed on CaCO_3_ Microparticles and the Release of a Protein Included by the Adsorption Method

**DOI:** 10.3390/polym12030520

**Published:** 2020-03-01

**Authors:** Egor V. Musin, Aleksandr L. Kim, Sergey A. Tikhonenko

**Affiliations:** Institute of Theoretical and Experimental Biophysics Russian Academy of Science, Institutskaya st., 3, Puschino, 142290 Moscow, Russia; eglork@gmail.com (E.V.M.); kimerzent@gmail.com (A.L.K.)

**Keywords:** polyelectrolyte microcapsules, dissociation, polyallylamine, polystyrene sulfonate, polyelectrolytes

## Abstract

The degradation of polyelectrolyte microcapsules formed on protein-free CaCO_3_ particles consisting of polyallylamine (PAH) and polystyrene sulfonate (PSS) and the resulting yield of protein in the presence of various salts of different concentrations, as well as at two pH values, was studied by fluorescence spectroscopy; the protein was incorporated into prepared microcapsules by adsorption. It was found that a high concentration of sodium chloride (2 M) leads to considerable dissociation of PAH, which is apparently due to the loosening of polyelectrolytes under the action of ionic strength. At the same time, 0.2 M sodium chloride and ammonium sulfate of the same ionic strength (0.1 M) exert less influence on the amount of dissociated polymer. In the case of ammonium sulfate (0.1 M), the effect is due to the competitive binding of sulfate anions to the amino groups of the polyelectrolyte. However, unlike microcapsules formed on CaCO_3_ particles containing protein, the dissociation of polyelectrolyte from microcapsules formed on protein-free particles increased with increasing temperature. Apparently, a similar effect is associated with the absence of a distinct shell, which was observed on microcapsules formed on protein-containing CaCO_3_ particles. The high level of the presence of Fluorescein isothiocyanate (FITC)-labeled Bovine Serum Albumin (BSA) in the supernatant is explained by the large amount of electrostatically bound protein and the absence of a shell that prevents the release of the protein from the microcapsules. In 2 M NaCl, during the observation period, the amount of the released protein did not exceed 70% of the total protein content in the capsules, in control samples, this value does not exceed 8%, which indicates the predominantly electrostatic nature of protein retention in capsules formed on protein-free CaCO_3_ particles. The increase in protein yield and peeling of PAH with increasing pH is explained by the proximity of pH 7 to the point of charge exchange of the amino group of polyelectrolyte, as a result of the dissociation of the microcapsule.

## 1. Introduction

Polyelectrolyte microcapsules (PMC), produced by the method of alternately layering oppositely charged polyelectrolytes on microsized dispersed particles, with subsequent destruction and removal of these particles, are the objects of a new rapidly developing field–polymer nanotechnology [1,2,3,4,5].

The results obtained thus far demonstrate the wide possibilities of using PMC in the development of a new class of chemical and biochemical reactors and the study of the physical and chemical processes in a small volume when creating a new type of probes and highly sensitive sensors and developing original methods for separating mixtures of various organic and inorganic substances in particular, the separation of heavy metal ions from the medium [4,6,7,8,9,10]. Along with such works, which have a pronounced technical orientation, at present several researchers are actively developing the PMC in relation to the development of long-acting drugs with controlled delivery and controlled release [11,12,13,14,15]. To successfully solve such problems, it is necessary to know the dynamics of the release of encapsulated substances and the degradation of the shell.

There are several articles describing the release of low molecular weight drugs from polysaccharide microcapsules. Thus, it was shown in [16,17,18,19] that multilayer microcapsules, whose shell has different thickness and composition, provide effective prolonged release of water-soluble drugs such as furosemide, ibuprofen and dexamethasone. In addition, the literature describes the yield of high molecular weight substances. The influences of the pH of the medium and the thickness of the capsule shell on the yield of insulin are described in [20,21]. The release of bovine serum albumin from polyelectrolyte microcapsules consisting of dextran sulfate and polyarginine, as well as from alginate and polyarginine, is described in article [22]. Using the example of tetramethylrhodamine isothiocyanate (TRITC)-labeled bovine serum albumin, it is shown that its yield from microcapsules depends on the number of polyelectrolyte layers of the membrane, the volume of the solution, and also the method of loading the protein into microcapsules [23].

The effect of temperature and the presence of salts on the rearrangement of polyelectrolyte groups is being actively studied. For example, possible mechanisms of influence are demonstrated by B. Schlenof et al. [24]. It was shown that the lifetime of the polycation–polyanion complexes was inversely proportional to the concentration of NaCl. C. Helm et al. demonstrated at article the dynamics of dissociation of polystyrene sulfonate (PSS) from layer-by-layer (LbL) films under the influence of NaCl was studied and, in particular, showed the polycation degree of polymerization is much larger than the polyanion degree of polymerization, PSS becomes largely immobilized within the multilayer film [25]. In the work of Daniel F. Kienle and Daniel K. Schwartz, a microscopic single-molecular transport of fluorescently labeled polylysine in polyelectrolyte layers was observed under the influence of NaCl of various concentrations and complex salt-dependent changes in the viscoelastic properties of the films, which balance the intermolecular binding and molecular conformation, are considered [26].

In addition, Dubrovskii A.V. and co-authors studied the degradation of the shell of polyelectrolyte microcapsules consisting of polyallylamine (PAH) and polystyrene sulfonate (PSS) and the yield of protein from them in the presence of NaCl and (NH_4_)_2_SO_4_ in different concentrations, and at pH 5 and 7.4. These microcapsules were prepared by alternately adsorbing the oppositely charged polyelectrolytes on a compound spherulite CaCO_3_ protein (Figure 1 below). Such microspherulites were obtained by the ion exchange reaction when mixing calcium chloride and sodium carbonate in the presence of protein in solution. It was shown that a high concentration of sodium chloride (2 M) leads to a marked dissociation of PAH from the upper layer of the shell, which is due to its loosening under the influence of ionic strength. Less than 20% of the polymer leaves in a solution of 2 M NaCl and only 2% of the polymer is released in water. An increase in the pH of the solution to 7.5 also causes peeling of PAH, however this effect decreases approximately by a factor of 2 with an increase in temperature to 37 °C due to the ordering and compaction of the envelope, which is shown by electron microscopic studies. In addition, it showed that the encapsulated protein practically does not leave the microcapsules, regardless of the presence of salts, their concentration and pH in the medium [27]. However, for some biologically active substances, such a procedure for inclusion in microcapsules cannot be used, due to aggressive pH values during the formation of spherulites, the possibility of elution of metal ions from the molecules of EDTA metal proteins, when dissolving them, etc. In this connection, other methods of protein incorporation are used, in particular, absorbing into the already formed microcapsules.

Kazakova Li with co-authors showed that the microcapsules formed on the protein-free CaCO_3_ core have a complex intravascular structure that remains after its removal. However, PMCs formed on the protein-containing CaCO_3_ nucleus completely lack this complex structure—polyelectrolytes are located only in the surface layer and there is a formed shell [28]. This feature of the structure and intrastructural organization of microcapsules can influence the dynamics of dissociation in different types of capsules. To encapsulate the protein in polyelectrolyte microcapsules, the adsorption method is used, obtained on protein-free CaCO_3_ particles, which consists of the incubation of microcapsules in the protein solution [28]. 

The purpose of this work is to study the dissociation dynamics of a shell of polyelectrolyte microcapsules, the structure of which is arranged in sponge-type (spongy type) consisting of polyallylamine (PAH) and polystyrene sulfonate (PSS) and the resulting yield of protein.

## 2. Materials and Methods

### 2.1. Materials

Polyelectrolytes polystyrenesulfonate sodium (PSS) and polyallylamine hydrochloride (PAH) with a molecular mass of 70 kDa, fluorescein isothiocyanate (FITC) Mw = 389.38, bovine serum albumin (BSA), Ethylenediaminetetraacetic acid disodium salt dihydrate (EDTA) purchased in Sigma (St. Louis, MO, USA), sodium chloride, ammonium sulfate, sodium carbonate, calcium chloride from “Reahim”.

### 2.2. Preparation of Fluorescently Labeled PAH and BSA

To a solution of polyelectrolyte or protein (10 mg/mL) in 50 mM borate buffer, pH 9.0 with stirring (300–400 rpm), FITC was slowly added in the same buffer. The components were fused in a molar ratio of FITC:PAH (BSA) (by amino groups) = 1:100. After 1.5–2 h of incubation, the resulting solution was dialyzed against water (10 L) overnight.

### 2.3. Preparation of CaCO_3_ Microspherulites

To 0.33 M CaCl_2_ solution, vigorously agitated on a magnetic stirrer, an equal volume of 0.33 M Na_2_CO_3_ solution was rapidly added. Stirring was continued for 30 s. The resulting suspension was maintained until complete precipitation of the formed particles. The process of “ripening” of microspherulites was controlled with the help of a light microscope. Then, the supernatant was decanted, the precipitate was washed with water and used to prepare PMC. The microparticles were obtained with a narrow size distribution with an average diameter of 5 μm.

### 2.4. Preparation of Polyelectrolyte Microcapsules

Seven-layer polyelectrolyte microcapsules were obtained by alternately adsorbing the oppositely charged polyelectrolytes onto a dispersed microspherulites (core), followed by dissolution of this cores [13]. Alternate adsorption of PSS and PAH on the surface of CaCO_3_ microspherulites was carried out in solutions of polyelectrolytes with a concentration of 2 mg/mL containing 0.5 M NaCl. Each step of adsorption was followed by a triple wash with a 0.5 M NaCl solution, which was necessary to remove unadsorbed polymer molecules. The particles were separated from the supernatant by centrifugation. After applying the required number of layers, the carbonate kernels were dissolved in a 0.2 M EDTA solution for 12 h. The resulting capsules were washed three times with water to remove nuclear decay products. Capsules containing labeled PAH as the 7th (outer) layer were obtained.

### 2.5. Inclusion of Protein in the Microcapsule by the Adsorption Method

Polyelectrolyte microcapsules were incubated in a solution of BSA protein labeled with FITC (6 mg/mL) for 12 h. The resulting microcapsules were washed three times with bidistilled water to remove unadsorbed protein molecules. Registration of the results of inclusion was made by comparing the fluorescence of the protein solution before and after switching on.

### 2.6. Registration of the Dissociation of the Polyelectrolyte Microcapsules and the Yield of Protein from Polyelectrolyte Microcapsules

The dissociation of the microcapsule and the yield of the protein from them were studied by fluorescence spectroscopy. Polyelectrolyte microcapsules, containing FITC-labeled PAH, into which the FITC-labeled BSA was encapsulated, were precipitated by centrifugation at 3000 rpm for 1 min. Further, the fluorescence of the supernatant was measured. Fluorescence spectra were recorded on an Infinite 200 Tecan instrument in a thermostated cuvette with an optical path length of 1 cm when excited with light at a wavelength of 273 nm.

## 3. Results and Discussion

Figure 1 shows plots of the mass of the FITC-labeled PAH in the supernatant from the incubation time of polyelectrolyte microcapsules at room temperature (22 °C) (Figure 1a) and at 37 °C (Figure 1b). From the figure, it can be seen that the presence of various salts in different mass in the solution has a significant effect on the dissociation of polyelectrolytes of microcapsules. Thus, the effect of 2 M sodium chloride on the microcapsules is quite strong, as can be seen from the significant increase in the amount of polymer in the solution. At the same time, 0.2 M sodium chloride and ammonium sulfate of the same ionic strength (0.1 M) exert less influence on the dynamics of dissociation of the polymer and the amount of dissociated polymer. This can be explained by the fact that at such a high salt concentration the microcapsules loosen because of the screening effect of ions on the sulfo and amino groups of the polyelectrolytes. Earlier, we found that ammonium sulfate removes the effect of interaction of PAH with urease even at a concentration of 0.01 M [29,30]. This is supposedly related to the formation of a sulfate-PAH bond, since one anion of sulfate can be contacted with at least two amides of PAH, which should lead to an increase in the rigidity of the polymer chain of PAH and, accordingly, a decrease in the ability to complement it in the most important regions of the molecule for catalysis squirrel. However, as can be seen from Figure 1, in the case of an interpolyelectrolyte PAH-PSS complex, such a mechanism does not work and as a consequence, PAH does not appear in significant amounts in solution. In this connection, it can be concluded that if the salt affects the dissociation of polyelectrolytes, this is not due to the competitive binding of the sulfo groups of the salt to the amino groups of the polyelectrolyte, but to a purely electrostatic nature. 

In our previous work [27], the dissociation of a different type of capsules—capsules formed on protein-containing CaCO_3_-spherulites—was studied. A decrease in the dissociation of polyelectrolytes with increasing temperature was also demonstrated. The effect was due to the fact that as the temperature is increased, the amount of ionic groups increases in “complementary” segments of polyelectrolyte chains coupled together by the electrostatic interaction and hydrogen bonds between oppositely charged ionic groups structural units. As a result, the contribution of the electrostatic repulsion decreases and the dispersion attraction between chain segments on the inner surface increases. This leads to an enhancement of hydrophobic interactions and, correspondingly, to a “pulling” of polymer chains inward, toward the center of the capsule and decrease the amount of polyelectrolytes in incubation solution [31]. In the case of sponge-like capsules, an increase in the dissociation of microcapsules is observed, however, the temperature does not exert a significant influence on the dissociation process. This may be due to the fact that spongy-type capsules do not have a clearly defined shell structure and, as a consequence, there is no compaction (ordering of unrelated polyelectrolyte sites), which prevents the dissociation of microcapsules. A slight increase in dissociation is associated with an increase in the mobility of polyelectrolyte molecules with increasing temperature.

Further, the destruction of the shell of polyelectrolyte microcapsules as a function of the pH of the medium was studied. In Figure 2 shows the dependence of the mass of FITC-labeled PAH in the supernatant of the incubation time polyelectrolyte microcapsules at room temperature (22 °C) and at 37 °C at pH 5 and 7. It can be seen that the greatest amount of PAH and, consequently, the degree of destruction of the shell, observed at pH 7 at a temperature of 37 °C. This behavior of envelope may be explained by the fact that the pH value is close to the point of recharge amino, which leads to a weakening of connections between –NH_3_^+^ and –SO_3_^-^ groups polyelectrolytes (pKa of PAH is 8.8 [32]; pKa of PSS is 1.22 [33]). When studying the capsules obtained in CaCO_3_-spherulites containing protein, the highest level reached polymer dissociation at 22 °C in a solution at pH 7, at 37 °C capsules become more resistant to degradation [27]. In the case of capsules spongy type of effect is observed, on the contrary, an increase of the medium temperature increases dissociation of the polymer dynamics, as in the case of the salts.

The next stage was the study of the dynamics of protein released from polyelectrolyte microcapsules. In Figure 3 shows plots of the amount of protein released from the microcapsules as a function of their incubation time at room temperature (22 °C) (Figure 3a) and at 37 °C (Figure 3b). Figure 3a,b shows that in the presence of 2 M sodium chloride, the amount of BSA increased in the solution in the course of the first 10 h with a further outlet to the plateau. In 2 M NaCl, the amount of the released protein did not exceed 70% (700 μg) of the total protein content in the capsules, in control, this value does not exceed 8%, which allows to talk about the predominantly electrostatic nature of protein retention in spongy type capsules.

The results of the study of the yield of BSA from polyelectrolyte microcapsules as a function of the pH of the medium are shown in Figure 4. It follows from figure an increase in the release of BSA from microcapsules at pH 7 at both temperatures is evident. This effect is explained by the proximity of pH 7 to the point of charge exchange of the amino group of the polyelectrolyte as a result of the disintegration of the microcapsule.

## 4. Conclusions

In this paper, a study was conducted to study the dissociation of polyelectrolyte microcapsules formed on protein-free CaCO_3_ particles and the yield of a protein encapsulated by adsorption. It was found that a high concentration of sodium chloride (2 M) leads to considerable dissociation of PAH, which is apparently due to the loosening of polyelectrolytes under the action of ionic strength. At the same time, 0.2 M sodium chloride and ammonium sulfate of the same ionic strength (0.1 M) exert less influence on the amount of dissociated polymer. However, unlike microcapsules formed on CaCO_3_ particles containing protein, the dissociation of polyelectrolyte from microcapsules on protein-free particles increased with increasing temperature. A similar effect is apparently associated with the absence of a distinct shell, which was observed on microcapsules formed on protein-containing CaCO_3_ particles. The high level of the presence of FITC-labeled BSA in the supernatant is explained by the large amount of electrostatically bound protein and the absence of the envelope bounding the protein from the external environment. In 2 M NaCl, during the observation period, the amount of protein released did not exceed 70% of the total protein content in the capsules, in control, this value does not exceed 8%, which allows to speak about the predominantly electrostatic nature of protein retention in sponge type capsules. The increase in protein yield and dissociation of PAH with increasing pH is explained by the proximity of pH 7 to the point of charge exchange of the amino group of the polyelectrolyte as a result of the disintegration of the microcapsule.

## Figures and Tables

**Figure 1 polymers-12-00520-f001:**
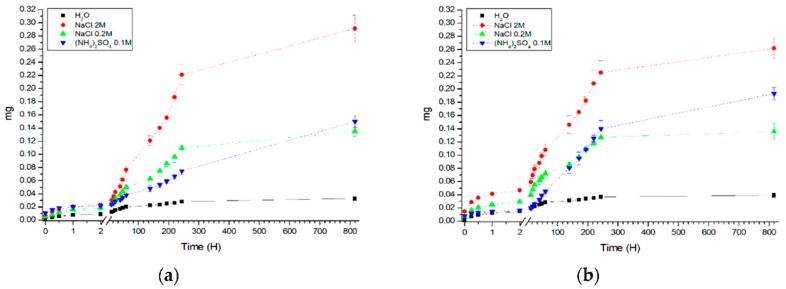
Dependence of the mass of the FITC-labeled PAH in the supernatant on the incubation time of polyelectrolyte microcapsules at room temperature (22 °C) (**a**) and at 37 °C (**b**) in the presence of salts.

**Figure 2 polymers-12-00520-f002:**
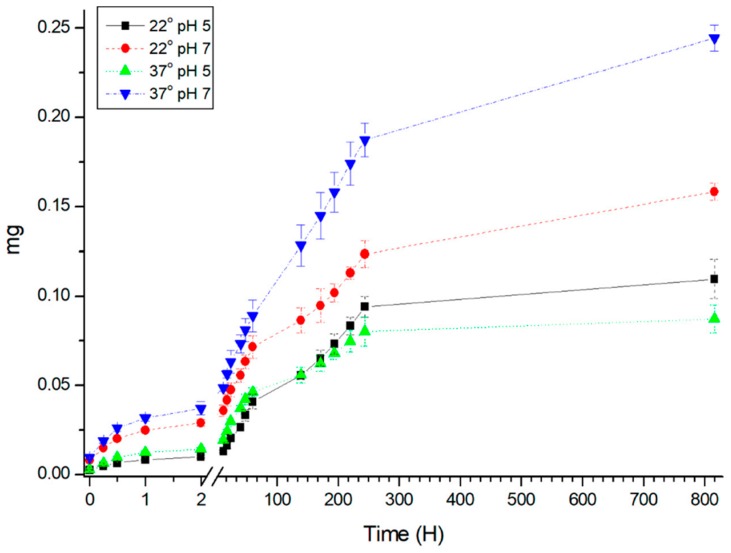
Dependence of the mass of the FITC-labeled PAH in the supernatant versus time incubation of polyelectrolyte microcapsules, the envelope of which contains FITC-labeled PAH at room temperature (22 °C) and at 37 °C, at pH 5 and 7.4. 1—pH 5, 22 °C; 2—pH 7.4, 22 °C; 3—pH 5, 37 °C; 2—pH 7.4, 37 °C.

**Figure 3 polymers-12-00520-f003:**
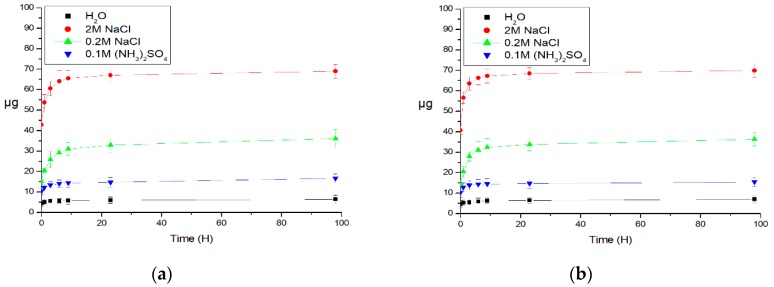
Dependence of the mass of FITC-labeled BSA in the supernatant on the incubation time of polyelectrolyte microcapsules, at room temperature (22 °C) (**a**) and at 37 °C (**b**) in the presence of salts.

**Figure 4 polymers-12-00520-f004:**
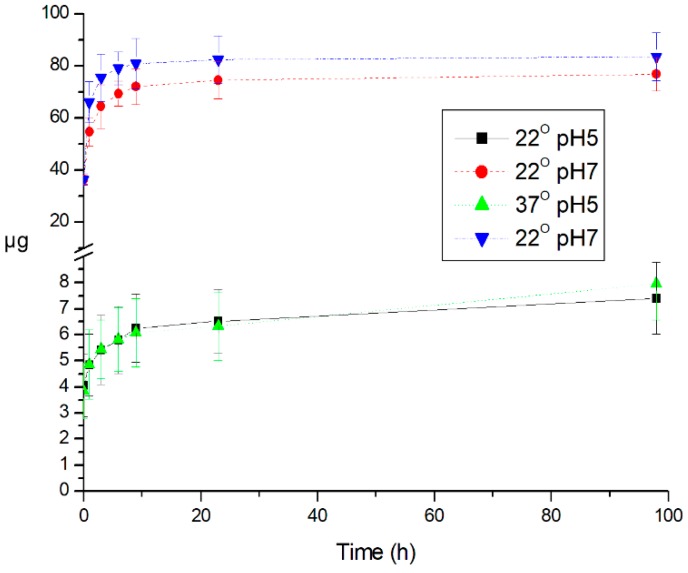
Dependence of the mass of FITC-labeled BSA in the supernatant on the incubation time of polyelectrolyte microcapsules, at room temperature (22 °C) and at 37 °C, at pH 5 and 7.

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
