# Peer review of "Destruction of Polyelectrolyte Microcapsules Formed on CaCO3 Microparticles and the Release of a Protein Included by the Adsorption Method"

_polymers, 2020, doi:10.3390/polym12030520_

Round 1

Reviewer 1 Report

Fig. 1 . Scale bars on A,C,D does not show the actual scale
Section 2.1 what is the PDI of polymers? Mw of FITC?
158, page 5. How authors distinguish the difference in the release kinetics related to the temperature change to the explanation provided in the manuscript?
The comparison of two graphs does not show the significant effect of the temperature and the salt type on the release. The NaCl effect is quite similar.
The last paragraph on the page 5 - during the discussion of the pH effects and polyelectrolyte ionization it is helpful to provide pKa values of PAH and PSS, how this values are related to the effects the authors observe?
188 rephrase the sentence
Fig 5. What is the "mcg" units? Please use IS notation and format.
The description of the Fluorescence apparatus is missing

Recent references are missing - the only one citation is dated by 2019 when the other groups lead by Schlenoff, Sukhishvili, Kozlovskaya, Helm, and others are not even mentioned. These groups have many publications last year regarding the salt effects on the release of proteins and other model molecules.

Author Response

Dear Reviewer, We thank you for your review and hope that we answered all your questions.

List of questions and answers:

  • Fig. 1 . Scale bars on A,C,D does not show the actual scale

That figure deleted.

  • Section 2.1 what is the PDI of polymers? Mw of FITC?

We provide Mw of FITC in the text. PAH and PAS Mw from 30kD to 90kD, the percentage of monomers is less than 10%. PDI is not specified by the manufacturer.

  • 158, page 5. How authors distinguish the difference in the release kinetics related to the temperature change to the explanation provided in the manuscript?

The effect was due to the fact that As the temperature is increased, the amount of ionic groups increases in “complementary” segments of polyelectrolyte chains coupled together by the electrostatic interaction and hydrogen bonds between oppositely charged ionic groups structural units. As a result, the contribution of the electrostatic repulsion decreases and the dispersion attraction between chain segments on the inner surface increases. This leads to an enhancement of hydrophobic interactions and, correspondingly, to a “pulling” of polymer chains inward, toward the center of the capsule and decrease the amount of polyelectrolytes in incubation solution.

  • The comparison of two graphs does not show the significant effect of the temperature and the salt type on the release. The NaCl effect is quite similar.

Yes, it is, this information is indicated in the text of the work: "In the case of sponge-like capsules, an increase in the dissociation of microcapsules is observed, however, the temperature does not exert a significant influence on the dissociation process."

  • The last paragraph on the page 5 - during the discussion of the pH effects and polyelectrolyte ionization it is helpful to provide pKa values of PAH and PSS, how this values are related to the effects the authors observe?

We provide pKa values of PAH and PSS in the text. Incubation of PMC at the solution with pH = 7 leads to a weakening of connections between -NH3+ and –SO3- groups polyelectrolytes, because of recharge amino groups of PAH.

  • 188 rephrase the sentence

The sentence was rephrased.

  • Fig 5. What is the "mcg" units? Please use IS notation and format.

We changed mcg to μg.

Reviewer 2 Report

This is an interesting work. This manuscript described the study of the influence factor of dissociation dynamics of microcapsule shells composed of polyelectrolytes, including polyallylamine hydrochloride (PAH) and polystyrene sulfonate (PSS), and protein release from the shells. The adsorption method was utilized innovatively in the procedure of inclusion of protein to facilitate the release of protein, which overcomes the problem that the encapsulated protein practically does not leave the microcapsules in the current study.

However, it still remains some concerns.

  1. I am curious to check the reason of choosing 2 M NaCl as disassociation medium, as this concentration is quite high. And high salt concentration might lead to undesired protein un-stability. And which scenery could present the case of 2 M NaCl, in terms of protein release from microcapsule?
  2. For Figure 1, it might not be proper to put the reference data in the figure of main text, as it is not a review. By the way, it needs permission from ref 22 before use.
  3. I am interested in the structure of microcapsule. Can I check whether the protein is combined with microcapsule by electrostatic reaction or inside the capsule? Please provide the experimental evidence. Furthermore, can you explain the link between PAH and protein release, as shown in Figure 2 and Figure 3.
  4. For Figure 4 and Figure 5, the y axis should be stated as ug, rather than mcg for easy understanding.
  5. As the encapsulation method changes, I am curious about the structure of the microcapsule. It would be better if the TEM or SEM studies will be conducted.
  6. Please explain more on the value of studying salt medium and pH on the disassociation of microsphere, in terms of protein release.

I am also attaching some specific comment:

(1) Page 2, Line 54. There should be “are” instead of “is”.

(2) Page 2, Line 79. There should be “showed” instead of “it was shown”.

(3) Page 4, Line 138. The word “In” at the beginning of paragraph should be deleted.

Author Response

Dear Reviewer, We thank you for your review and hope, that we answered all your questions.

List of questions and answers:

  • I am curious to check the reason of choosing 2 M NaCl as disassociation medium, as this concentration is quite high. And high salt concentration might lead to undesired protein un-stability. And which scenery could present the case of 2 M NaCl, in terms of protein release from microcapsule?

PMC have electrostatic nature and for that reason, we chose 2M NaCl, which increases ionic strength of the medium. That can lead to PMC structure changes and to their destruction. Also, the contribution of electrostatic repulsion can be increased.

  • For Figure 1, it might not be proper to put the reference data in the figure of main text, as it is not a review. By the way, it needs permission from ref 22 before use.

Figure 1 deleted and ref 22 was added to the text.

  • I am interested in the structure of microcapsule. Can I check whether the protein is combined with microcapsule by electrostatic reaction or inside the capsule? Please provide the experimental evidence. Furthermore, can you explain the link between PAH and protein release, as shown in Figure 2 and Figure 3.

The link between PAH and protein release is absent. Dissociation of polyelectrolyte reflect the integrity of PMC structure, but protein release reflects the electrostatic nature of protein encapsulation from that kind of polyelectrolyte microcapsules.

  • For Figure 4 and Figure 5, the y axis should be stated as ug, rather than mcg for easy understanding.

We changed mcg to μg.

  • As the encapsulation method changes, I am curious about the structure of the microcapsule. It would be better if the TEM or SEM studies will be conducted.

TEM and SEM will not allow assessing the dissociation of single polyelectrolyte molecules from PMC. Dissociation is not so pronounced that it can be seen in TEM and SEM.

  • Please explain more on the value of studying salt medium and pH on the disassociation of microsphere, in terms of protein release.

We supposed that protein encapsulated because of electrostatic nature. For that reason, we chose conditions of medium with different ionic strength and pH. That conditions allow us to show clearly the nature of encapsulation of that kind of microcapsules.

  • I am also attaching some specific comment

Text corrected based on these comments

Round 2

Reviewer 2 Report

There is still one concern. With removal of Figure 1, the authors should re-organize the figure number, instead of starting from Figure 2. 

Author Response

Dear Reviewer, we thank you for noticing our mistake in the numbering of pictures and giving us the opportunity to improve the article.

Numbering the figures is now in the correct sequence.

With all regards, Dr. Tikhonenko.